# AN ANALYSIS OF QUERY-BASED APPROACHES TO IMAGE ATTRIBUTE PREDICTION

**Aaron Walsman, Daniel Gordon, Dieter Fox**
Paul G. Allen School of Computer Science and Engineering
University of Washington
Seattle, WA 98102, USA
{awalsman,danielgordon,fox}@cs.washington.edu

## ABSTRACT

Deep learning techniques have emerged as the de facto method for solving many classification-based computer vision tasks (He et al. (2016); Ren et al. (2015); Krause et al. (2015)). Each of these tasks require multiple (often hundreds of) examples of each category in order to learn accurate classifiers. More recently, complex visual reasoning tasks have been proposed to challenge this classification-based paradigm (Johnson et al. (2017)). Deep networks that succeed on the CLEVR task learn to combine information from multiple sub-systems rather than attempting to extract all necessary information in a single forward pass (Santoro et al. (2017)). We explore a similar setting which compares multi-class classification networks against query-based networks across a wide variety of attributes in a single image. We show that query-based networks outperform traditional multi-class networks given a fixed network capacity due to their ability to focus on information relevant to the current query. We also show that query networks learn faster than multi-class networks because their focus-based representation on specific attributes allows for more multi-modal flexibility per training iteration.

## 1 INTRODUCTION

Much of the power of deep learning stems from its ability to encode generic features which can be combined hierarchically in order to form complex decision boundaries which generalize well to new examples. CNNs are often described as extracting different granularities of image features (edges, textures, parts) and combining them in order to perform classification (Zeiler & Fergus (2014)). However, this style of network cannot be easily extended to tasks with more complex output spaces such as attribute prediction and visual question answering (VQA). A naïve classification network for attributes (e.g. one class for white dog, one for black dog etc.) would have exponentially more output classes than a typical classification network . A multi-class prediction network (e.g. one output for white or not, one for dog or not etc.) would force the network to encode many disparate image features into an overlapping feature space. Similarly, a VQA network which feeds the question in after extracting generic image features forces the network to encode enough information to potentially answer *any* question that could be asked about the image. This fundamental problem will appear in any network which requires a single representation to be reused for multiple unrelated decisions. To mitigate this issue, we explore a simpler form of **query networks**, a model which is inherently multi-modal and can represent different information based on the various decisions that must be made.

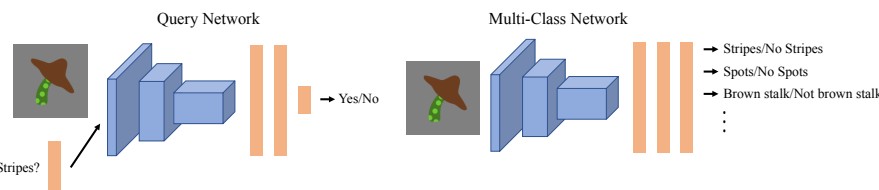

Figure 1: Comparison of query network structure to traditional multi-class classification networks.

## 2    QUERY NETWORKS

Query networks (shown in Figure 1) take as input both an image and a query about certain properties of the image. This allows the network to switch its focus and disentangle multi-modal problems much more directly. In this work, the queries take the form of an attribute one-hot encoding, but they could be extended to more complex encodings such as question embeddings or texture patch encodings. Others have suggested similar architectures, but normally query embeddings are connected to fully-connected layers, meaning all visual processing has already occurred (Antol et al. (2015); Santoro et al. (2017)). Our structure allows the network to adjust visual feature extraction based on the query itself. In our experiments, we explore the implications of this structure on a controlled task in order to show its effectiveness; we show that query networks with limited capacity outperform traditional multi-class networks.

## 3    EXPERIMENTAL FRAMEWORK

We design a testing framework which directly compares multi-class classification networks with query networks. We generate data in a randomized fashion which simplifies the feature learning problem and removes the possibility of unintentional image bias (i.e. no attribute will be easier or harder by chance based on some unrelated image features). We also control for network capacity by using the same underlying structure for multi-class and query networks and by testing across a range of architectures.

### 3.1    DATA

In order to explore the differences between query-based and query-free learning we built a synthetic dataset designed to produce simple images with many attributes. These images contain cartoons of mushrooms with various sizes, shapes, colors and patterns. Examples of these images can be seen in Figure 2. Each image is generated by randomly choosing several discrete parameters such as cap style, color, and pattern, and real-valued parameters such as height, width, and bend angle. Once the image is generated, these parameters are discretized into thirty-five binary values that serve as a label vector for the image. This dataset has five thousand training images and five thousand test images, each with a 35-bit label vector. Each image is $128 \times 128$. The goal of the learning system is to predict yes or no for each of the 35 attributes. We consider each attribute's accuracy independently in order to avoid biasing our results on performance on more difficult attributes.

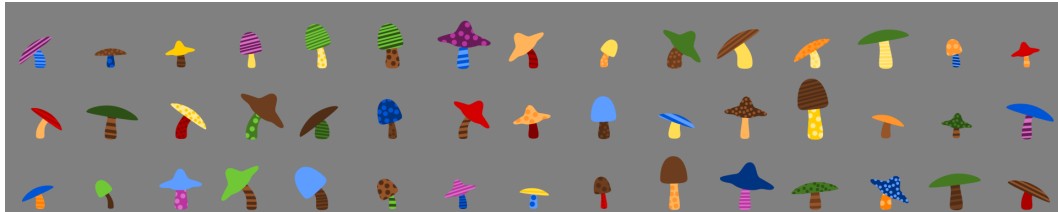

Figure 2: Forty-five example images from our dataset. The top left mushroom has attribute labels "blue stalk, purple cap, thin stripes, left tilt" among others. Best viewed in color.

### 3.2    NETWORK STRUCTURES

We directly compare performance of query networks with multi-class networks by keeping their overall structure and capacity similar for each experiment. We additionally train multiple independent single-attribute networks to provide an upper bound on performance given a fixed structure. We conducted several ablation tests to study the effect of different network parameters on performance (such as number of convolutional layers), but due to space considerations we report only the ablation on network capacity, which had the greatest effect on overall performance. Therefore all networks in this paper consist of four convolutional layers followed by three fully-connected layers. Query networks have an additional input of the query attribute index which is fed through an embedding layer and tiled to $128 \times 128$. This tiling is stacked on top of the image and fed into the first convolutional layer. All convolutional layers have kernels of size $3 \times 3$ and are followed by ReLU nonlinearities, batch normalization Ioffe & Szegedy (2015) and $2 \times 2$ max-pooling with a stride of 2 for downsampling. All fully-connected layers have ReLU nonlinearities and dropout of 0.5 during training. In order to test performance on our dataset, we trained each network for forty epochs. We use the Adam optimizer (Kingma & Ba (2014)) with a fixed learning rate of 1e-4 and a batch size of 64 for all experiments.

## 4    RESULTS

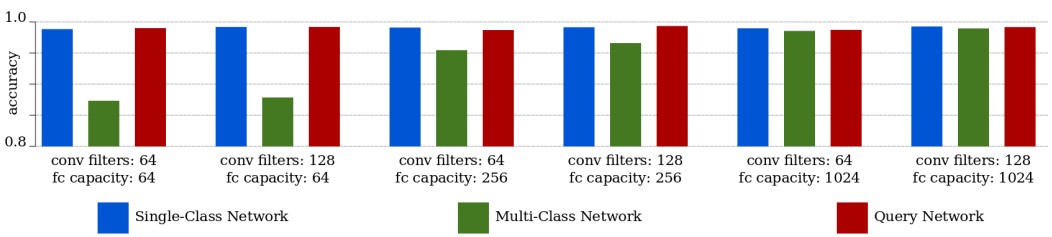

Figure 3: Bar charts showing the performance of various network architectures on the Mushroom test set. We average accuracy across all attributes to avoid biasing results based on the hardest classes. Network capacity increases to the right.

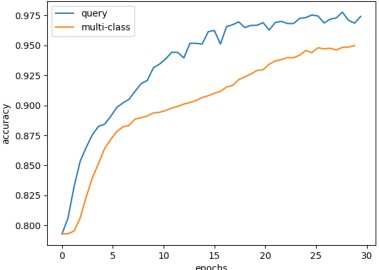

Figure 4: Test set performance during training. The query networks learn much faster than the equivalent multi-class networks. Here we show the networks with 64 convolution filters and 256 fully connected features.

We first evaluate the performance of various network architectures on the Mushroom dataset, reporting accuracies for multiple models of varying network capacity in figure 3. Traditional multi-class classification networks struggle to learn feature extractors at limited capacity, as shown by the first two plots in figure 3. With equivalent capacity, query networks are much more able to master the Mushroom task. For single-class networks, we train individual networks on each feature, allowing them to fully specialize on a specific attribute. These act as an upper bound on network performance for a given capacity. The query networks match the single-class networks' performance in nearly all cases.

By increasing the overall capacity of the networks, multi-class networks can match the performance of query networks. This indicates that query networks are effectively able to switch modes and reuse existing capacity based on the input attribute, whereas multi-class networks are forced to represent all features simultaneously.

Figure 4 shows the performance of the query and multi-class networks during training. Query networks are able to learn much faster than multi-class networks, likely because the gradients of different inputs will not directly compete to adjust the same network weights; because the query network can extract different features based on the inputs, the outputs (and gradients) can be uncorrelated between two questions. This is not the case for multi-class networks which must use the same feature representation to answer multiple unrelated questions.

## 5    CONCLUSION

In this paper, we evaluate the effectiveness of query networks compared to traditional multi-class networks, showing an improvement in performance and training time for a fixed network capacity. This suggests that the query networks can adjust their focus based on the inputs rather than being forced to represent all aspects of the image in a single feature extraction step. We believe query networks can increase performance in many tasks where the output depends on more than just the image itself.

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
