# OpenReview forum: "An Analysis of Query-Based Approaches to Image Attribute Prediction"
_ICLR.cc/2018/Workshop — Reject_

### Official Review · AnonReviewer2 · 2018-03-10
**Interesting experiment but lack of novelty and detailed analysis**

**Rating:** 4
**Confidence:** 4

**Review:**

This paper explores query-based approaches to image attribute prediction. The authors proposed a simple query network and compare with traditional multi-class classification network on a synthetic dataset with many attributes. The authors compare with different network structures with similar capacity and find the query network is able to switch modes and reuse existing capacity based on the attributes. However, this paper is lack of novelty and detailed analysis of the experiment.

1) The query-based network is not novel since most previous VQA or other X+QA method can belong to this category. The proposed query network is just a simple baseline of those methods.

2) The authors didn't mention the split of the datasets, and why use the synthetic, not real attribute image dataset?

3) The experiment analysis shows that query network can match single class network on overall performance with similar parameters, while with more and more parameters, the multi-class network can gradually match the performance. This is interesting, however, the detailed analysis is missing, which might give more insight of the experiment.

---

### Official Review · AnonReviewer1 · 2018-03-11
**Interesting idea, but limited experiments and missing citations**

**Rating:** 6
**Confidence:** 4

**Review:**

SUMMARY
The authors explore query networks as an alternative approach to multi-class classification. Rather than outputting a distribution over all possible classes, a query network receives as input the category it should predict, and it outputs a decision for that category alone. Experiments on a synthetic dataset demonstrate that query networks outperform multi-class networks when using smaller network architectures, and perform on-par with multiclass networks as network size increases.

Pros:
(+) Clean, simple idea
(+) Paper is clearly written
(+) Mushroom dataset is interesting

Cons:
(-) Similar to answer embeddings which have previously used for VQA
(-) Some missing citations
(-) Limited experiments
(-) No discussion of test-time computational complexity

NOVELTY
Query networks seem like a nice way to deal with complex output spaces; the notion that this allows the network to adapt its feature extraction to the particular category under consideration is intuitive. These query networks are similar to the answer embeddings used in state-of-the-art methods for VQA including [1] and [2] (which are not cited); however query networks are slightly different since they inject class information into the early CNN layers while answer embeddings for VQA inject this information into fully-connected layers.

EXPERIMENTS
The experiments on the synthetic mushroom dataset show that the idea has merit, but I would have liked to see a bit more experimental evaluation. I appreciate the ablation study of different network sizes, but I would have also liked to see results on some standard image classification datasets such as CIFAR10 or CIFAR100. I would also like to see experimental validation of the claim that query networks can “adjust visual feature extraction based on the query itself”; to test this claim the authors could for example compare with a version of the model where class information is injected into the fully-connected layers.

The mushroom dataset also seems like an interesting contribution on its own - will this dataset be publicly released?

COMPUTATIONAL COMPLEXITY
One downside of query networks compared to multiclass networks is that query networks will require C independant forward passes to fully annotate an image with C potential classes, while a multiclass network requires only a single forward pass. I would have liked to see some discussion of this potential downside.

OVERALL
On the whole query networks seem like an interesting idea, and the paper presents some preliminary results demonstrating that the idea may be worthy of further investigation. The novelty and experiments are somewhat lacking, but may be sufficient for a workshop paper.

REFERENCES
[1] Jabri et al, “Revisiting Visual Question Answering Baselines”, ECCV 2016
[2] Fukui et al, “Multimodal Compact Bilinear Pooling for Visual Question Answering and Visual Grounding”, EMNLP 2016

---

### Official Review · AnonReviewer3 · 2018-03-13
**Potentially Interesting Idea**

**Rating:** 5
**Confidence:** 4

**Review:**

In this work, the authors present controlled synthetic experiments comparing standard multiclass convolutional neural nets against query-based versions which output a binary decision given an input query (\ie cap shape) and image. This experiment seems to indicate that the query networks perform substantially better than standard networks at lower capacity.

One point of contention for me is the claim that the networks have the same capacity. Considering the fact that the one-hot query representation is tiled across the input map, the initial set of filters are something on the order of 3x3x(35+3) parameters compared to the 3x3x3 in the standard net.  Notably, only one set will be active at a time such during any forward pass only 9 extra weights can be active per spatial location. Furthermore, the primary point of restriction demonstrated in the results (Figure 3) is changing the width of the fully connected layers. This seems somewhat trivial that the multiclass network will do worse with fewer neurons given that same bottleneck needs to encode information about every attribute while the query network must simply encode the binary result.

In general, I think this paper is on the right track and has the potential to provide valuable insight; however, this workshop submission may be a bit premature.

---

### Decision · Program_Chairs · 2018-03-20
**ICLR 2018 Workshop Acceptance Decision**

**Decision:**

Reject

**Comment:**

Based on the reviews, this paper has not been accepted for presentation at the ICLR workshop. However, the conversation and updates can continue to appear here on OpenReview.